# Quality of life and associated factors among the youth with substance use in Northwest Ethiopia: Using structural equation modeling

**Gebrekidan Ewnetu Tarekegn** [1] *, **Goshu Nenko** [2], **Sewbesew Yitayih Tilahun** [2], **Tilahun Kassew** [2], **Demeke Demilew** [2], **Mohammed Oumer** [3], **Kassahun Alemu** [1], **Yassin Mohammed Yesuf** [4], **Berhanie Getnet** [2], **Mamaru Melkam** [2], **Eden Abetu Mehari** [5], **Biruk Fanta Alemayehu** [2] *

1 Department of Epidemiology and Biostatistics, Institute of Public Health, College of Medicine and Health Sciences, University of Gondar, Gondar, Ethiopia, 2 Department of Psychiatry, College of Medicine and Health Sciences, University of Gondar, Gondar, Ethiopia, 3 Department of Human Anatomy, School of Medicine, College of Medicine and Health Sciences, University of Gondar, Gondar, Amhara, Ethiopia, 4 Department of Psychology, College of Social Science and Humanities, University of Gondar, Gondar, Amhara, Ethiopia, 5 Department of Clinical Pharmacy, School of Pharmacy, College of Medicine and Health Sciences, University of Gondar, Gondar, Ethiopia

* fantabiruk@yahoo.com (BFA); ewnetuwendale@gmail.com (GET)

**Data Availability Statement:** All relevant data are within the paper and its Supporting Information files.

## Abstract

### Background

Substance use leads to serious clinical conditions with the potential to cause major health and emotional impairments in individuals. Individuals with substance use typically report significantly poorer QoL than the general population and as low as those with other serious psychiatric disorders. It has a high impact on morbidity, mortality, and productivity, it also compromises the general safety and performance of the users, i.e., affects the quality of life. Therefore, this study aimed to assess quality of life and identify the potential predictors among youths who use substances.

### Methods

A multicenter cross-sectional study design was applied to assess quality of life and associated factors among substance use youths in the central Gondar zone from January 1 to March 30/ 2021. A total of 373 substance use youths were included in the study. The data were collected using face-to-face interview by structured questionnaires, and entered to Epi-data version 4.6 and exported to STATA version 16, and AMOS software for further statistical analysis. To identify factors associated with health-related quality of life, structural equation modeling was used, and it also used to estimate the relationships among exogenous, mediating, and endogenous variables.

### Results

Substance used youths had a moderate overall health-related quality of life (mean score = 50.21 and 14.32 standard deviation, p-value < 0.,0001), and poor health-related quality of

**Funding:** This study was funded by the University of Gondar with reference number VP/RCS/003/2013. The datasets supporting the conclusions of this manuscript are available and uploaded as supported materials.

**Competing interests:** The authors have declared that no competing interests exist.

**Abbreviations:** GFI, Goodness of Fit Index; CFA, Confirmatory Factor Analysis; CFI, Comparative Fit Index, NFI: Normal Fit Index; MSPSS, Multidimensional Scale of Perceived Social Support, HRQOL: Health-Related Quality of Life; SEM, Structural Equation Modeling; RMSEA, Root Mean Square Error of Approximation; SD, Standard Deviation; SRQ-24, Self-Reporting Questionnaire, TLI: Tucker-Lewis Index; WHO, World Health Organization; WHOQOL, World Health Organization Quality of Life.

life in the environmental health domain (mean score of 45.76 with standard deviation of 17.60). Age ($\beta$ = 0.06, p<0.001), sex ($\beta$ = 0.30, p<0.001), psychotic symptoms ($\beta$ = -0.12, p<0.001), employment status ($\beta$ = 0.06, p = 0.008,), loss of family ($\beta$ = 0.35, p<0.001), and social support ($\beta$ = 0.27, p<0.001) were variables significantly associated with health-related quality of life.

## Conclusion

According to the findings of this study, substance abuse during adolescence is associated with lower health-related quality of life and a higher report of psychopathological symptoms. Given this finding, mental health and health promotion professionals should learn about and emphasize the impact of substance use on youth quality of life.

## Background

Substance use has become one of the major public health issues with its pervasive prevalence, wide spread across all categories of society; according to the Global Addiction 2017 report, around 1in 5 to1 in 20 individuals aged 15 years old reported using heavy alcohol, tobacco and illicit drugs daily in past month [1]. Substance-Related Disorders are among the most common social problems caused by using legal and illegal substances [2]. A systematic review on alcohol use prevalence in Eastern Africa found 52% and 15% over use and problem use prevalence respectively [3]. A systematic review of substance abuse in our country also show significantly higher rate of substance use, i.e., alcohol, Khat, tobacco, etc. at-risk populations, including youth, compared with the general population [4]. Another systematic analysis on prevalence of lifetime substances use among students, high school and university, showed that the lifetime prevalence of any substance use was 52.5% (95% CI 42.4–62.4%), and that of Khat 24.7% (95% CI 21.8–27.7%), alcohol 46.2% (95% CI 40.3–52.2%), and smoking cigarette 14.7% (95% CI 11.3–18.5%) [5].

Substance use related disorders are associated with a significant disease burden and the highest mortality among all mental and behavioral disorders, for example, with five times higher mortality compared to the general population in alcohol use disorders [3]. The leading cause of accidental injury and death (e.g., automobile accidents, suicide) among adolescents is precipitated by substance use. Substance related problems also cause significant economic impact from issues like lost productivity and lives and health care costs [6].

Substance use, in addition to the mentioned impact on morbidity, mortality, and productivity, it also compromises the general safety and performance of the users, i.e., affects the quality of life (QoL).

In addition to assessing the impact of certain variables in terms of morbidity, mortality, and economic costs, there is a growing trend of assessing societal progress by measuring 'quality of life', beyond economic growth indicators like GDP [7]. Even if it is not a universally agreed definition, the WHO defines quality of life as "an individual's perception of their position in life in the context of the culture and the value systems in which they live and in relation to their goals, expectations, standards and concerns [8].

Quality of life is a broad concept and is affected by a number of factors; from independent demographic factors such as age and sex to other interacting factors such as level of one's physical health, socioeconomic status such as level of education, marital status and others social relationships, living area and economic independence, psychological state such as presence or absence of depression, personal beliefs [8–15].

Different literature in different settings shows substance use is the major factor associated with that the quality of life of youth segment of the society in general [2,16].

A cross-sectional study done on the relationship between alcohol/drug use and quality of life among adolescents that included 5 countries, India, Indonesia, Nigeria, Serbia, Turkey, Bulgaria, and Croatia, found that alcohol/drug use was significantly associated with lower levels of QOL [16].

A study done in Iran on the quality of life of adolescents and young people who arrived at addiction treatment centers showed the total average score of quality of life was low at admission and showed significant improvement after treatment [17].

A study on quality of life among individuals entering substance use disorder treatment in Norway found that approximately three-fourths of both genders self-reported their QoL as "very poor" or "poor", and 25% rated it "neutral" or higher. There were no significant differences in the distribution of women and men's QoL [18].

Another exploratory study on one-year outcomes done in Norway on quality of life and substance use disorders found that the majority reported remarkably low QoL. Using a single item to measure overall QoL, 75.8% (414) reported their QoL as "very poor" or "poor", 17.8% (97) as neutral, and 6% (33) as "good", and only 0.4% (2) as "very good" [3].

The other factors that appeared to affect quality of life in youth, particularly those who uses substance, were socioeconomic variables. A population-based longitudinal study done on psychosocial risk and protective factors of child and adolescent health-related quality of life in German, showed that low socioeconomic status and migration background were both associated with low health-related quality of life while self-efficacy, family climate, and social support were positively associated with initial health related quality of life [19].

Research done on young individuals in Spain and Iran, to examine if educational level has an influence on health-related quality of life found that that the higher the level of education, the better the level of health-related quality of life [2,20].

In a study done in South Korea in patients with alcohol use disorder, which includes young individuals, we found that socioeconomic factors such as stable income, stable employment, stable residence, and social support directly and indirectly affect quality of life, and a similar study also has pointed out the negative impact of the psychological state, particularly depression on the quality of youth who uses alcohol [21].

There also has been found an association between physical health and quality of life. In two studies, which included young participants, done on cardiac patients in Ethiopia and cystic fibrosis in Canada found that living with such conditions is associated with poor overall health related quality [11,22].

By assessing quality of life, we can identify groups with poor quality of life, and this could guide interventions that will improve their situation and avert more serious consequences, allocate limited resources based on unmet needs, guide a strategic plan, and monitor the intervention.

However, the literature review showed that there are limited studies conducted on health-related quality of life among substance use youth in general in Africa, specifically in Ethiopia, in particular. To address the above gaps, we conduct this study with sufficient sample size and appropriate statistical analysis by testing the following hypothesis, a) null hypothesis = substance user youths have good health-related quality of life, alternative hypothesis = substance user youths have poor health-related quality of life, b) alternative hypothesis = health related quality of life is associated with socio-demographic, family-related, and psychosocial factors.

## Methods and materials

### Study setting and period

Amhara regional state, Northwest Ethiopia, and covers an area of 21791.83 km2. It is a newly established zone that was previously located within the North Gondar zone. This zone is divided into 16 districts (15 rural districts and one special district) and 442 kebeles. According to 2020 population estimates, the Central Gondar Zone population is 2,642,138 people. Youths (15–24 years old) made up 575,656 (21.79 percent) of the population, with 286,385 and 289,271 males and females, respectively. Nowadays, youths are heavily exposed to the use of various substances. This reduces the effort of the youth in terms of productivity, mentality, and quality of life. As a result, assessing the quality-of-life substance youths is critical in order to have a productive power of youths.

### Population

All youths (15–24 years old) who are living in Central Gondar zone were a source population, and all youths in central Gondar zone who are living in the selected "*kebeles*" of the zone were selected as the study participants. However, youth who are unable to communicate due to severe mental/ physical illness during the data collection period were excluded from the study.

### Sample size determination

For structural equation modeling, Nunnally suggested that a ratio of 10 cases per variable would be a sufficient sample when latent variables have multiple indicators [23]. Since in this study each latent variables have more than six indicators, and the total number of measured variables includes 26 indicators, 11 independent variables that give 37 total observed variables. Therefore, by the ongoing rule of thumb, the adequate sample sizes to estimate the parameters were 370.

### Sampling techniques and procedures

Since the study area is broad, which is difficult to address all parts of the zone in a single step. The study employed multistage sampling technique. We stand in central Gondar zone and selected 6 districts, then a number of '*Kebeles'* are also randomly selected from these districts. There are also special centers '*Got'* inside Kebele. Here we have recruited our participants from the Got in the form of cluster by considering a proportional allocated sample in each kebeles.

### Variables and measurements

Data were collected using face-to-face interview by structured questionnaires. The questionnaire consists of five parts such as: socio-demographic characteristics, family-related questions, WHO-QOL, social support related, and psychotic symptoms questions.

The socio-demographic characteristic and family related questions were developed based on the previous literatures [2,19–21]. Whereas the standard questionaries were used for the measurements of health-related quality of life, social support, and psychotic symptoms.

The Health-related quality measurement domain questionnaire was adopted from WHO-QOL which is also validated in Ethiopia and other parts of the countries in the world [24]. It has 4 domains that denote an individual's perception of quality of life in each particular domain. The WHOQOL-BREF is a 26-item instrument consisting of four domains: physical health domain (7 items), psychological health domain (6 items), smjfocial relationships

domain (3 items), and environmental health domain (8 items); it also contains the overall perception of QOL and general health (2 items).

The social support was measured using the multidimensional Scale of Perceived Social Support (MSPSS) tool. The tool is designed to measure perceived social support from three sources: Family, Friends, and other. The scale is comprised of a total of 12 items, with 4 items for each subscale [25]. Each item response is a Likert scale from 1 (very strongly disagree) to 7 (very strongly agree).

Self-Reporting Questionnaire (SRQ-24) also was used to assess psychotic symptoms of the youths. SRQ-24 is an instrument with 24 items which question respondents about symptoms and problems, 20 related to neurotic symptoms, and 4 items concerning psychotic symptoms. This study interested to use SRQ-4 consisting only of the "psychotic" items for the assessment of psychotic symptoms at the community level. Each of the 4 items is scored 0 or 1. A score 1 indicates that the symptom was present during the past 30 days, a score of 0 indicates that the symptom was absent, the maximum score is 4. Individuals with the total sum score 2 or more were considered as having psychotic symptoms [26].

## Data processing, model building and analysis

After the data were collected, the data were coded and entered into Epi-data version 4.6, and then exported to STATA version 16 and Amos version 25 for further analysis. Descriptive and summary statistics were done using figures and tables. common method variance (CMV) was investigated by Harman's Single-Factor Test. The result shows that the first unrotated factor captured only 28% of the variance in data. Thus, the two underlying assumptions did not meet, i.e., no single factor emerged and the first factor did not capture most of the variance. Therefore, these results suggested that CMV is not an issue in this study. To check the internal consistency of the tool, Cronbach's α was analyzed for each domain of WHO-QOL–Brief. The values of Cronbach's α coefficient with 0.7 or higher were considered satisfactory [27]. The score of each domain of WHO-QOL–Brief was obtained by averaging their corresponding items for each participant [28].

The SEM was employed to examine the relationship between various exogenous and endogenous or mediated variables. The parameter in the model was estimated using the maximum likelihood estimation method. The analysis was started with the hypothesized model (Fig 1), and modifications were performed iteratively by adding path links or including mediator independent variables. Root Mean Square Error of Approximation (RMSEA), Comparative Fit Index (CFI) was calculated to assess the goodness of fit of the given model. model with value RMSEA < 0.05 and CFI > = 0.95 was retained. Diagrammatically, the effect of each exogenous or mediating variable on the respective latent variable was indicated by the path coefficient along with a single headed arrow, and the correlation among disturbances was indicated by double arrows. When mediation of effects was present, the direct, indirect, and total effects were calculated.

## Ethical consideration

Ethical clearance was obtained from Institutional Review Board (IRB) of University of Gondar and an official permission letter was gained from each selected "*wereda*". First, aim of the study was explained verbally to the participants and after their willingness, written permission was obtained from the study participants before filling the questionnaire. For individuals less than 18 years old, written informed assent was obtained from their parents. Confidentiality was maintained by omitting their personal identification. Participants was not being forced to participate and received any monetary incentive, and it was solely voluntary based.

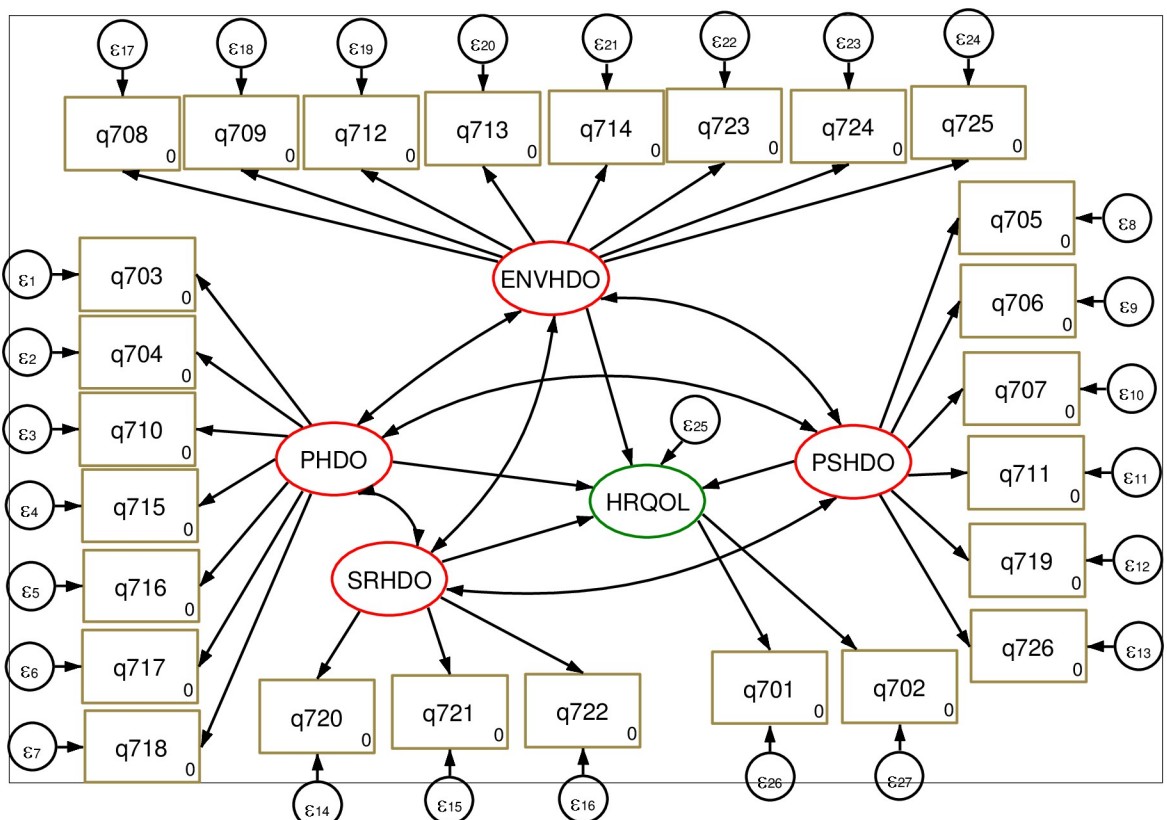

**Fig 1. Hypothesized Path diagram of health-related quality of WHOHRQOL-Breff developed from the literatures [29].** Where: HRQOL: Health-related quality of life; PHD: Physical health domain; ENVHD: Environmental health domain, SRHD: Social relations domain, PSHD: Psychological health domain; q701:overall QOL, q702:overall health; q703:Pain and discomfort; q704:Medical treatment dependence; q705: Energy and fatigue; q706: Mobility; q707: Sleep and rest; q708: Daily activity; q709: Working capacity; q710: Positive feeling; q711: Spirituality/personal beliefs; q712: Memory and concentration; q713: Bodily image and appearance; q714: Self-esteem; q715: Negative feelings; q716: Personal relationships; q717: Sex; q718: Social support; q719: Physical safety and security; q720: Physical environment; q721, financial resources; q722: Information and skills; q723: Recreation and leisure; q724: Home environment; q725: Health accessibility and quality; q726: Transport.

## Results

### Socio-demographical and family-related characteristics of the substance used youths

Three hundred seventy-two (372) youths were willing and interviewed the questionnaire with a response rate of 97.8%. Of the total respondents, 114 (30.65%) were from Gondar zuria district, 274 (73.66%) were male, 316 (84.95%) were orthodox Christian, 184 (49.46%) achieved Secondary school and 310 (83.00%) are single, 217(58.3%) are students in occupation, 266 (71.51%) were lived with their family, 308(82.80%) were there biological parents alive, 108 (29.035%) of them was loss their family recently, and 222(59.685) were live in urban. The mean age of substance youths was 20.51 (2.61 SD) years (Table 1).

### Internal consistency & correlations between the domains of the WHOQOL-BREF

Cronbach's alpha was calculated for each domain of the instrument to check the internal consistency. All domains of WHOQOL-BREF had high values of Cronbach's alpha ($\alpha > 0.7$).

**Table 1. Socio-demographic and family-related characteristics of the respondents.**

| Variables | Categories | Frequency(N = 372) | Percept (%) |
|---|---|---|---|
| District | Gondar | 45 | 12.0 |
| | Alefa | 79 | 33.33 |
| | Chilga | 62 | 16.67 |
| | East Belesa | 54 | 14.52 |
| | Tach Armachiho | 18 | 4.84 |
| | Gondar zuria | 114 | 30.65 |
| Religion | Orthodox | 316 | 84.95 |
| | Muslim | 48 | 12.90 |
| | Protestant | 6 | 1.61 |
| | others | 2 | 0.54 |
| Living with | Alone | 95 | 25.54 |
| | With family | 266 | 71.51 |
| | Others | 11 | 2.96 |
| Biological parents alive | Yes | 308 | 82.80 |
| | no | 64 | 17.20 |
| Sex | Male | 274 | 73.66 |
| | Female | 98 | 26.34 |
| Residency | Urban | 222 | 59.68 |
| | Rural | 150 | 40.32 |
| Marital status | Single | 310 | 83.33 |
| | Married | 55 | 14.78 |
| | Divorced/separated | 4 | 1.08 |
| | Windowed | 3 | 0.81 |
| Educational level | Cannot read and write | 31 | 8.33 |
| | Primary education | 79 | 21.24 |
| | Secondary education | 184 | 49.46 |
| | Collage and above | 78 | 20.97 |
| Occupation | Government employ | 30 | 8.06 |
| | Merchant | 55 | 14.78 |
| | Farmer | 19 | 5.11 |
| | student | 217 | 58.33 |
| | day laborer | 27 | 7.26 |
| | house wife | 7 | 1.88 |
| | Others | 17 | 4.57 |
| Resent loss of families | Yes | 109 | 29.30 |
| | No | 263 | 70.70 |

Inter-domain correlation showed that there was a statistically significant correlation between domains, there is a highly positive correlation between environmental health domain and psychological health domain ($r = 0.61$, $p<0.001$) and as compared with other domains, psychological health domain and social relation health domain had a relatively weak correlation between them ($r = 0.49$, $p<0.001$).

## HRQOL among youth who use substance

Among the domains of health-related quality of life, substance youths scored the highest and lowest mean HRQOL score in the physical health domain (58.00 with 16.14 SD), and the social

**Table 2. HRQOL of substance youths in central Gondar zone, 2021.**

| Domain | N | Minimum | maximum | Mean | SD |
|---|---|---|---|---|---|
| Physical health | 372 | 0.00 | 100.00 | 58.00 | 16.14 |
| Psychological health | 372 | 0.00 | 91.67 | 50.11 | 14.67 |
| Social relation | 372 | 0.00 | 100.00 | 47.33 | 21.39 |
| Environmental health | 372 | 0.00 | 100.00 | 45.76 | 17.60 |
| HRQOL | 722 | 7.14 | 88.69 | 50.21 | 14.32 |

Were, HRQOL: Health Related Quality of Life, SD: Standard deviation.

relation health domain (47.33 with 21.39 SD) that were moderate HEQOL. The mean score of overall HRQOL of substance youth was 50.21 (14.32SD) which was moderate HRQOL (Table 2).

### Perceived health satisfaction and self-rating of HRQOL of the respondent

Study participants were asked to give their perception of their quality of life and health satisfaction. Based on their response; about one-third, 147 (39.51%) youths reported that their quality of life was neither good nor poor, while 29 (7.80%) of them had very poor QOL. Regarding health satisfaction, 128 (34.41%) of them were dissatisfied with their health and only 23 (6.18%) of them were very satisfied with their health (Figs 2 and 3).

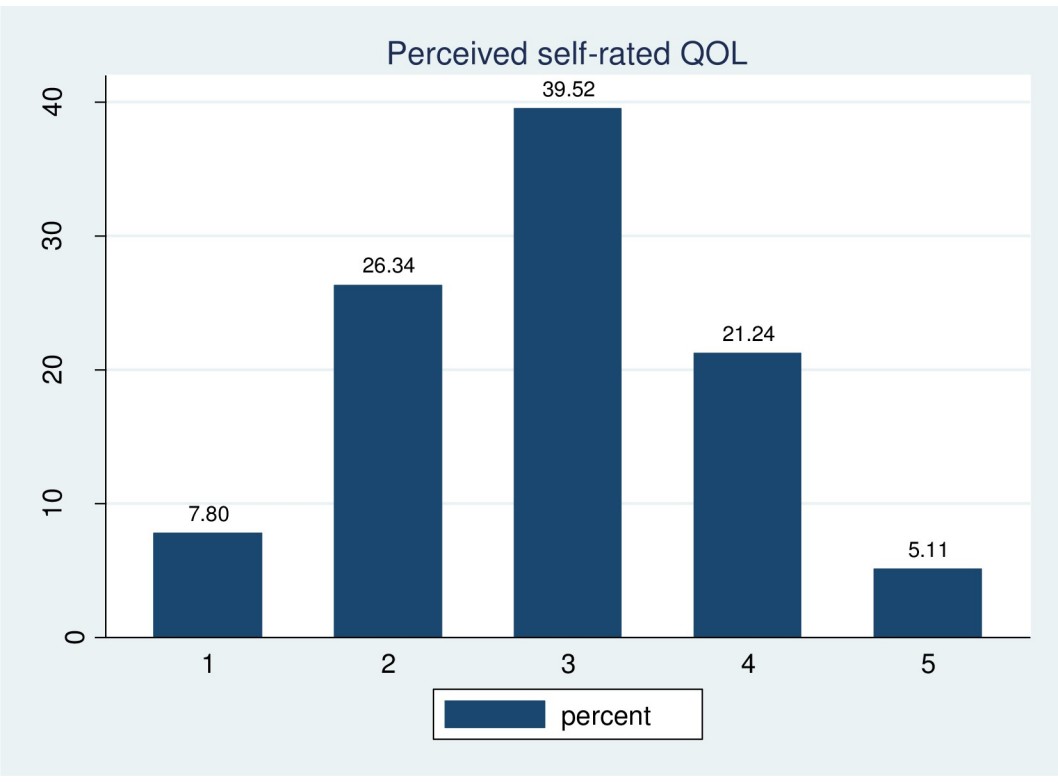

**Fig 2. Perceived self-rated QOL of substance youths in central Gondar zone, 2021.**

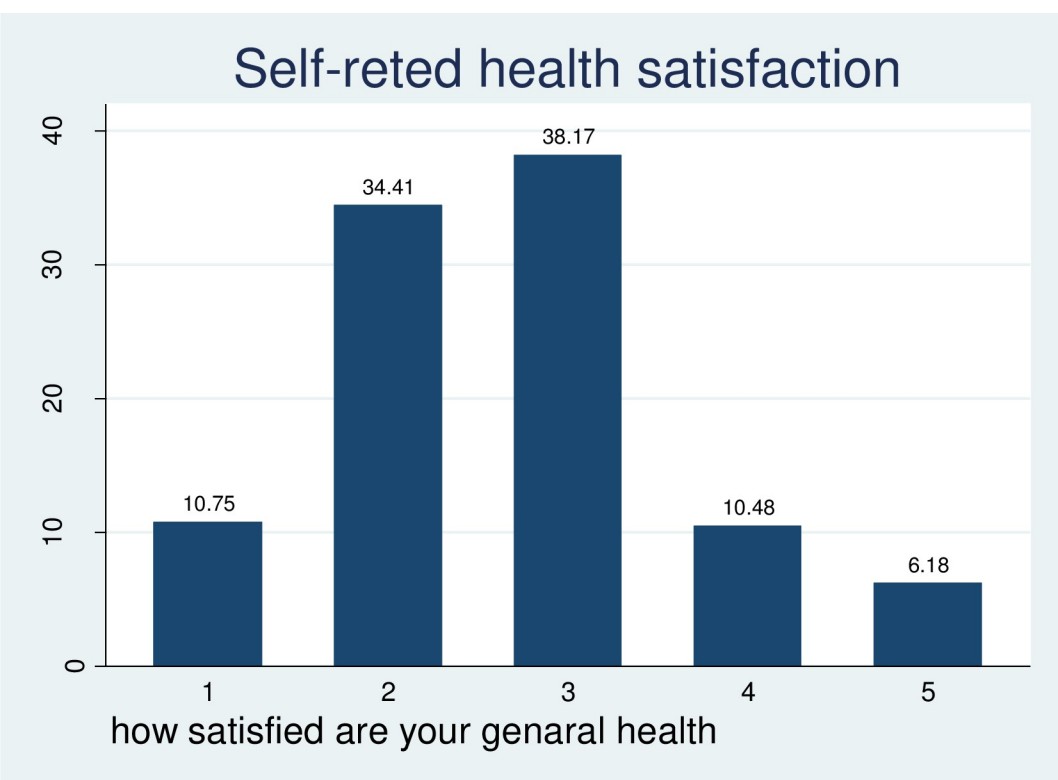

**Fig 3. Perceived self-rated health satisfaction of substances used by youths in central Gondar zone, 2021.**

## Factors associated with health-related quality of life among substance youths

The final model containing both the structural part (relationships between latent or observed variables) and the measurement part (relationship between a latent variable and its indicators or items) is shown in Fig 4 and Table 3. The fitted model was relatively parsimonious and well fitted with RMSEA = 0.03 and CFI = 0.90. Variables like residency, marital status, district, religion, living with, and parents alive were excluded from the final model as their contributions were not statistically significant at 5% of the level of significance.

In the fitted model, all path coefficients in the diagram were statistically significant at 5% of the level of significance. Consequently, the model included only seven exogenous variables (age, sex, loss of family, social support, psychotic symptoms, educational status, and having a job, four mediator variables (domains of HRQOL), and one endogenous variable (HRQOL). The exogenous variables (age, sex, loss of family, social support, psychotic symptoms, educational status, and having a job, three mediator variables (environmental health, psychological health, and social health domain) were, directly and indirectly, associated with HRQOL.

The structural equation model indicates that among the HRQO domains, psychological health factors had the most substantial effect on HRQOL, which was larger than the causal effect of environmental health factors on HRQOL, physical health, and social relationship factors was no significantly associated with HRQOL. We also assessed the effect of each socio-demographic and others variable on each domain of the HRQOL, that is, age (p<0.001), sex (p = 0.01), psychotic symptoms (p<0.001), loss of family (p<0.001), and social support (p<0.001) was significantly associated with psychological health domain, in the case of social relation domain, age (p<0.001), psychotic symptom(p<0.001), and social support(p = 0.001)

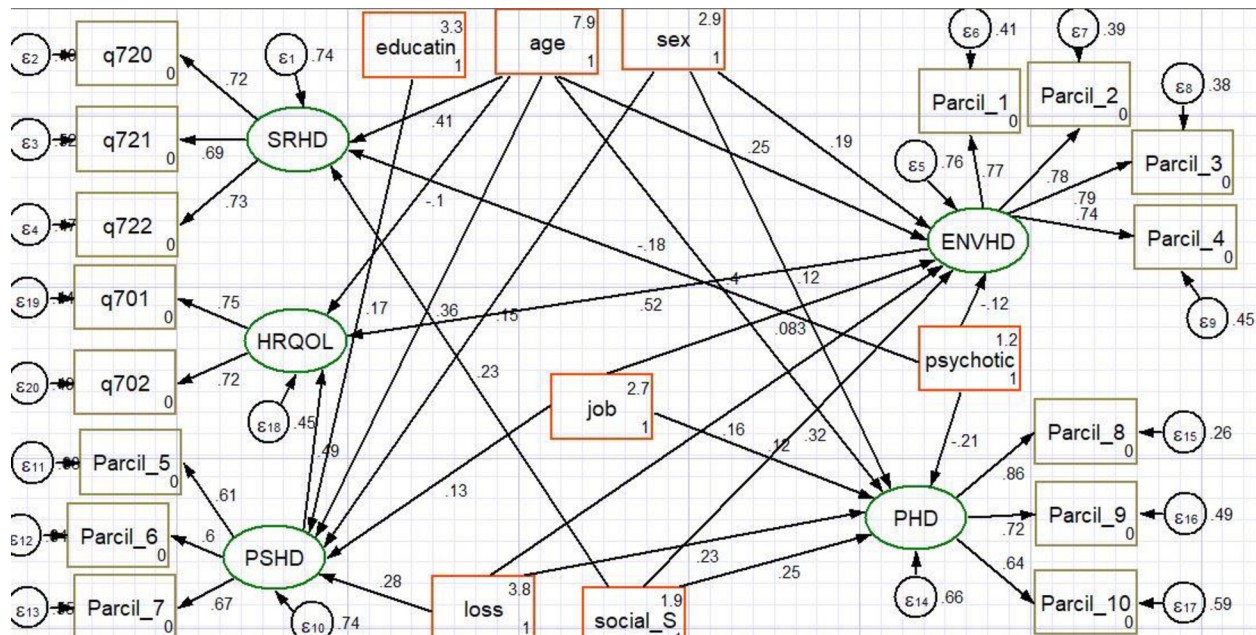

**Fig 4. SEM for factors associated with HRQOL for substance youths in central Gondar zone, 2021.** Where, PHD: Physical health domain, ENVHD: Environmental health domain, SRHD: Social relations domain, PSHD: Psychological health domain, parcil_1: Average of Q7 and Q13, parcil_2: Average of Q14 and Q24, parcil_3: Average of Q8 and Q23, parcil_4: Average of Q12 and Q25, parcil_5: Average of Q5 and Q11, parcil_6: Average of Q19 and Q7, parcil_7 = average of Q6 and Q26, parcil_8 = average of Q3, Q10 and Q17, parcil_9 = average of Q4 and Q16 parcil_10 = the average of Q18 and Q15, Resi: Residents of patients, education: Educational level of the youths, loss: Loss of beloved family, Social_S: Social support, psychotic: Psychotic symptoms of the youths, job: Job status of the youth.

was factors associated with it, while age, sex, job status, psychotic symptom, social support, and loss of family are factors associated with the environmental health domain, and variables like, educational status, age, sex, and job status was significantly associated with physical health domain (Fig 4).

## Test of the goodness of fit of the theoretical model

The results of the analysis of the structural equation model produced using the study variables in the hypothetical model were as follows: goodness of fit for GFI = 0.89, RMSEA = 0.03, NFI = 0.93, CFI = 0.90, TLI = 0.87, GFI indices satisfied the recommended levels.

## Effectiveness analysis of the hypothetical model

The direct, indirect, and total effects of the factors associated with the HRQOL of the youths are presented in Table 3. The psychological health domain had the greatest direct effect on the HRQOL with a score of 0.66 (95% CI of 041,0.91). The environmental health factor and the social relation health domain had no significant effect on the HRQO.L(p-value>0.05) at 95% level of confidence. Age had a direct effect on HRQOL with a path coefficient of -0.03, and a total effect of 0.06 when added to the indirect effect of environmental health domain and psychological health domain (Table 3).

## Discussion

A number of studies have been performed on HRQOL in older adults in the globe. This study is original and novel because it used a representative sample and a suitable statistical

**Table 3. The direct, indirect, and total effect of socio-demographical and clinical factors on HRQOL domains among youth who use substances.**

| Characteristics | | Direct Effect (95%CI), | Indirect effect (95%CI) | Total Effect (95%CI) |
|---|---|---|---|---|
| **DV: Physical health domain** | | | | |
| Age | | 0.11 (0.09, 0.12) | - | - |
| Sex | | | | |
| | female | 0 | 0 | 0 |
| | male | 0.19(0.04, 0.33) | - | - |
| Psychotic symptom | | | | |
| | No | 0 | 0 | 0 |
| | Yes | -0.12 (-0.17, -0.07) | - | - |
| Have Job | | | | |
| | No | 0 | 0 | 0 |
| | Yes | 0.06 (0.02, 0.11) | - | - |
| Social support | | | | |
| | Yes | 0.27 (0.17, 0.38) | - | - |
| | NO | 0 | 0 | 0 |
| Losses of family | | | | |
| | YES | 0.35(0.22, 0.49) | - | - |
| | No | 0 | 0 | 0 |
| **DV: Psychological health** | | | | |
| Sex | | | | |
| | Female | 0 | 0 | 0 |
| | Male | 0.20(0.06, 0.33) | - | - |
| | Age | 0.08(0.06, 0.09) | - | - |
| Having job | | | | |
| | YES | 0.05(0.01, 0.10) | - | - |
| | No | 0 | 0 | 0 |
| Losses of family | | | | |
| | Yes | 0.35(0.21, 0.48) | - | - |
| | NO | 0 | 0 | 0 |
| Education | | | | |
| | Illiterate | 0 | 0 | 0 |
| | Educated | 0.11(0.04, 0.19) | - | - |
| **DV: social relation** | | | | |
| Age | | 0.13(0.12, 0.15) | - | - |
| Psychotic symptom | | | | |
| Yes | | -0.12(-0.19, -0.05) | - | - |
| No | | 0 | 0 | 0 |
| Social support | | | | |
| | Yes | 0.30(0.16, 0.44) | - | - |
| | No | 0 | 0 | 0 |
| **DV: Environmental health domain** | | | | |
| Age | | 0.07(0.06, 0.09) | - | - |
| Sex | | | | |

*(Continued)*

**Table 3.** (Continued)

| Characteristics | Direct Effect (95%CI), | Indirect effect (95%CI) | Total Effect (95%CI) |
|---|---|---|---|
| **DV: Physical health domain** | | | |
| Female | 0 | 0 | 0 |
| Male | 0.32(0.16, 0.47) | - | - |
| Psychotic symptom | | | |
| Yes | -0.07(-0.13, -0.00) | - | - |
| No | 0 | 0 | 0 |
| Social support | | | |
| Yes | 0.37(0.26, 0.48) | - | - |
| No | 0 | 0 | 0 |
| DV: HRQOL | | | |
| ENHD | 0.54(0.37,0.71) | - | - |
| PSHD | 0.66(0.41, 0.91) | - | - |
| Age | -0.03(-0.06, -0.01) | 0.09(0.07, 0.11) | 0.06(0.04,0.08) |
| Sex | | | |
| female | 0 | 0 | 0 |
| male | - | 0.30(0.17,0,43) | - |
| Psychotic symptom | | | |
| No | 0 | 0 | 0 |
| Yes | -0.12 (-0.17, -0.07) | - | - |
| Have Job | | | |
| No | 0 | 0 | 0 |
| Yes | 0.06 (0.02, 0.11) | - | - |
| Social support | | | |
| Yes | 0.27 (0.17, 0.38) | - | - |
| No | 0 | 0 | 0 |
| Losses of family | | | |
| Yes | 0.35(0.22, 0.49) | - | - |
| No | 0 | 0 | 0 |

DV: Dependent variable, ENHD: Environmental health domain, PSHD: Psychological health domain, HRQOL: Health related quality of life.

methodology to provide information regarding the relationship between QOL and independent variables among substance youth.

In our study, we aimed to develop a theoretical model by reviewing different literature and verify the significance of the direct/indirect paths and the goodness of fit of the model under the theoretical assumption that demographic factors, personal related factors, family related factors, social relations, environmental factors, physical factors, psychotic symptoms of youths and behavioral factors, including depression, anxiety, fatigue, pain, sexual activity, and body image, determine the HRQOL of substance used youths.

In this study, we found that youths had moderate health-related quality of life in the overall mean score HRQOL (50.21 (14.32 SD)) and had lower quality of life in the environmental health domain. This finding is congruent with other previous studies [21,30]. The reasons for the experience of lower environmental health domain might be because of socio-demographic and cultural factors. In this study, the predominant participants were from a rural areas where there is lower personal and family income, shortage of recreational environment, poorer educational resources, and development technologies which are important to modify the youth health [31]. And it affects a large segment of the community that could pose challenges for

intervention. The current study has shown that many youths with substance use are suffering from poor quality of life due to the health effects of substance use in Ethiopia. Therefore, an appropriate environmental health promotion program for improving health-related quality of life is crucial for youth who use substances.

In this study, approximately 34.14% of the youths reported their quality of life as very poor and/or poor, and around 65.86% of their quality of life was neutral and/or above. This lower quality of life was observed because psychoactive substances resulted psychopathology like depression and anxiety symptoms, which causes impairment of quality of life in all health domains [32]. The lower score of quality of life could also be related to socio-demographic variables [33]. Unemployment and low income is predominant in the study area that results a low socio-economic status, in turn, the effects of poor health related quality of life experienced among the youth population. This finding is congruent with other previous studies [3,18]. This consistence may be due to; substance use is associated with significant emotional distresses and functional impairment, as explained in the literature review.

In our final model, environmental health factors like physical security, financial resources, and health care facility had the most substantial causal effect on HRQOL of substance youths with a path coefficient of 0.52 (95%CI, 0.37, 0.67). This is larger than the psychological health factors like: bodily image and appearance, negative feelings, positive feelings, self-esteem, spirituality / religion / personal beliefs, thinking, learning, memory, and concentration that had the most substantial causal effect on HRQOL of substance youths with a path coefficient of 0.49 (95%CI, 0.31, 0.68), which was larger than the causal effects in other domains. Our result is not supported by other studies done previously. This may be due to previous studies conducted in a developed countries and environmental health may not have a larger cause on HRQOL than the psychological health factors in developing countries like Ethiopia.

The effect of physical, environmental, social relations, and psychosocial health on overall health-related quality of life was assessed by confirmatory factor analysis, while the effects of sociodemographic, family-related, and psychometric related variable on each domain of HRQOL were evaluated using structural equation modeling simultaneously. Variables such as age, sex, psychotic symptoms, loss of family, and social support were significantly associated with the psychological health domain, while variables such as age, psychotic symptoms, and social support were significantly associated with the social relations domain. Variables such as age, sex, job status, psychotic symptoms, social support, and loss of family were factors significantly associated with the environmental health domain, while variables such as educational status, age, sex, and job status of the youth was significantly associated with in the physical health domain.

Age has a significant relation with each domain of HRQOL and has a negative direct and a positive indirect effect that resulted in a positive total effect on the overall HRQOL of substance use youth. As age increase, youth experience worsening of their mental state, physical health, and social relations. This result is congruent with a previous study done in the different setting [10,13]. This could be explained by the impact of a chronic substance on our brain, i.e., the alteration of its normal regulation on our appetite and ability to access healthy nutrition and withdrawal from social activities because of much time wasted in the process of using the substances, the loss of interest in social activities and the stigma from the environment towards those who use substances, the normal physiological changes with aging also could play role.

Educational status of the substance in youth was another variable associated with health-related quality of life, i.e., a better health-related quality of life is associated with a higher educational status. The finding is in line with study done Spain [20]. This may be a result of increasing health literacy and the influence of healthier life expectations in the society with higher educational achievement.

Results of the comparison of youths on HRQOL according to gender indicated significant differences between males and females on the quality-of-life dimensions, with females reporting lower than males on this variable. Studies on the adolescent's health-related QoL generally indicate that boys tend to report higher levels of life satisfaction, when compared to girls [34,35]. This may be due to the gender differences in the engagement of health-related behaviors, and it may be related to cultural and educational issues that assign different roles to males and females. However, we must consider the fact that males and females tend to vary in the perception of these dimensions, for example, women tend to show stronger emotional expressivity of their internal distresses.

Absence of social support from family and youths having psychotic symptoms are other predictors lowering the QOL of youths. This is supported by other studies [36,37]. This is may be because poor social support could precipitate negative emotional states which could lead the use of substances to self-medicate and vicious cycle of poor social support and more use which finally could result in poor quality of life. The presence of psychotic symptoms signifies the severity of the affection of their mental health which affect their quality of life.

## Strength and limitations

In the current study, QOL was measured using a standardized tool that is validated for both developed and developing countries. The study has been conducted in multicenter, and this may help to generalize the results to the population. The study also used appropriate statistical analysis to estimate the effect of different independent variables on several dependent variables and the subsequent direct comparison of the respective impact of the independent variables on the dependent variables. Nevertheless, this study is not without limitations, the study was conducted with small sample sizes to estimate the effect of predictors using SEM; further studies are needed to address this issue.

## Conclusions

The results of the present study suggest that substance abuse during adolescence is related to lower health-related QoL and a higher report of psychopathological symptoms. In light of this finding, mental health and health promotion professionals should learn about and magnify the impact of substance use on the quality of life of youths. Policy makers also should include education about substance in curriculums and take restraining measures on easy accessibility of substances. Future studies should assess quality of life in relation to specific psychopathological problems in such and even broader study populations.

## Supporting information

**S1 Data.**
(DTA)

## Acknowledgments

We would like to thank the University of Gondar for the approval of ethics to conduct this research. Additionally, we wish to express my sincere thanks and appreciation to data collectors and supervisors for their support during the data collection period.

## Author Contributions

**Conceptualization:** Gebrekidan Ewnetu Tarekegn, Goshu Nenko, Sewbesew Yitayih Tilahun, Tilahun Kassew, Demeke Demilew, Kassahun Alemu, Berhanie Getnet, Biruk Fanta Alemayehu.

**Data curation:** Goshu Nenko, Sewbesew Yitayih Tilahun, Mamaru Melkam, Eden Abetu Mehari.

**Formal analysis:** Gebrekidan Ewnetu Tarekegn, Tilahun Kassew, Demeke Demilew, Kassahun Alemu, Biruk Fanta Alemayehu.

**Investigation:** Goshu Nenko, Sewbesew Yitayih Tilahun, Tilahun Kassew, Demeke Demilew, Berhanie Getnet.

**Methodology:** Gebrekidan Ewnetu Tarekegn, Tilahun Kassew, Mohammed Oumer, Kassahun Alemu, Yassin Mohammed Yesuf, Biruk Fanta Alemayehu.

**Project administration:** Goshu Nenko.

**Resources:** Eden Abetu Mehari.

**Software:** Gebrekidan Ewnetu Tarekegn, Kassahun Alemu.

**Supervision:** Gebrekidan Ewnetu Tarekegn, Sewbesew Yitayih Tilahun, Tilahun Kassew, Demeke Demilew, Mohammed Oumer, Kassahun Alemu, Yassin Mohammed Yesuf, Berhanie Getnet, Eden Abetu Mehari.

**Validation:** Demeke Demilew, Yassin Mohammed Yesuf, Eden Abetu Mehari.

**Visualization:** Gebrekidan Ewnetu Tarekegn, Sewbesew Yitayih Tilahun, Tilahun Kassew, Mohammed Oumer, Yassin Mohammed Yesuf, Mamaru Melkam.

**Writing – original draft:** Gebrekidan Ewnetu Tarekegn, Kassahun Alemu, Biruk Fanta Alemayehu.

**Writing – review & editing:** Gebrekidan Ewnetu Tarekegn, Goshu Nenko, Sewbesew Yitayih Tilahun, Tilahun Kassew, Demeke Demilew, Mohammed Oumer, Kassahun Alemu, Yassin Mohammed Yesuf, Berhanie Getnet, Mamaru Melkam, Eden Abetu Mehari, Biruk Fanta Alemayehu.

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
