## [Decision Letter · Decision Letter 0]

22 Jul 2022

PONE-D-22-02286Quality of life and associated factors among youth who use substance, central Gondar zone, Northwest Ethiopia, 2021;  Using Structural Equation ModelingPLOS ONE

Dear Dr. Tarekegn

Thank you for submitting your manuscript to PLOS ONE. After careful consideration, we feel that it has merit but does not fully meet PLOS ONE’s publication criteria as it currently stands. Therefore, we invite you to submit a revised version of the manuscript that addresses the points raised during the review process.

We look forward to receiving your revised manuscript.

Kind regards,

Marianna Mazza

Academic Editor

PLOS ONE

“No funding”

Reviewers' comments:

Reviewer's Responses to Questions

**Comments to the Author**

1. Is the manuscript technically sound, and do the data support the conclusions?

Reviewer #1: Yes

Reviewer #2: No

2. Has the statistical analysis been performed appropriately and rigorously? 

Reviewer #1: Yes

Reviewer #2: Yes

3. Have the authors made all data underlying the findings in their manuscript fully available?

Reviewer #1: Yes

Reviewer #2: Yes

4. Is the manuscript presented in an intelligible fashion and written in standard English?

Reviewer #1: Yes

Reviewer #2: No

5. Review Comments to the Author

Reviewer #1: I hope some of these comments would be useful.

Abstract

1. please improve result section with apply and mention statistical.

2.How do they reach the conclusions in the abstract?

Background

1.please use update references in this section.

this references might be useful "Quality of life and its related factors in women with substance use disorders referring to substance abuse treatment centers" or "Can substance abuse media literacy increase prediction of drug use in students?"

2.whats are your hypotheses? please mentioned its in this section or methods section.

Methods

1.whats are response rate in your study?

2. sampling technique and study setting very vague. please explain more. Geographical cluster of each district, is setting of your study, or special centers of "Kebele" are your setting of your study?

3.Measurements

Were these questioner adapted from an existing tested questionnaire or did the authors develop it themselves, tested it?

Discussion

1.Discussion does not explain the findings, Authors state other studies, but do not offer an explanation as to why they see the results that they see

2.Please try to include more relevant literature.

Reviewer #2: The study sample size is reasonable with adequate rule of thumb for 370 subjects. The model appears to have reasonable goodness of fit and RMSEA values. The analysis software used, which is routine in this context, appeared to adequately identify factors associated with health-related quality of life which is a major factor, naturally. Structural equation modeling was used to estimate the relationships among exogenous, mediating, and endogenous variables which is notably the purpose of such software. There are many variables and the presentation is a descriptive summary of the relationships sought by the authors. One can get lost in the presentation with so much being described.

The characterization of the strength of the interdomain correlations is troubling. The authors note for example that Inter-domain correlation showed that there was a statistically significant correlation between domains, there is a highly positive correlation between environmental health domain and psychological health domain (r=0.61, p<0.001) and as compared with other domain's, psychological health domain and social relation health domain had a relatively

weak correlation between them (r= 0.49, p<0.001). How are weak and strong being decided?

The format of Table 3 makes it difficult to read and interpret the table. Please correct the alignment. There are typos in the manuscript. For example in the title of Fig. 3, Self Reted should obviously be Self Rated. Parcel in Figure 4 is in the footnote, but it looks like Parcil in the figure. The English should also be edited in places for clarity.

6. PLOS authors have the option to publish the peer review history of their article (what does this mean?). If published, this will include your full peer review and any attached files.

Reviewer #1: No

Reviewer #2: No

---

## [Author Response · Author response to Decision Letter 0]

11 Aug 2022

Author Response Letter

Manuscript ID: PONE-D-22-02286

Title: Quality of life and associated factors among substances user youths in Northwest Ethiopia: Using Structural Equation Modeling

Dear editor(s) and reviewers

First for all the authors would like to thank the editor(s) and reviewers for your precious time, thoughtful comments and constructive suggestions, which help to improve the quality of this manuscript. The corresponding changes and refinements made in the revised manuscript are summarized in our response below.

Q: reviewer’s comments, suggestions and questions

Response: authors response based on the editors/ reviewer questions and comments

Journal Requirements

Q1. Please ensure that your manuscript meets PLOS ONE's style requirements, including those for file naming. The PLOS ONE style templates can be found at

Response: To address the journal requirements, correction and revision have been made in the revised document.

Q2: Please include additional information regarding the survey or questionnaire used in the study and ensure that you have provided sufficient details that others could replicate the analyses. For instance, if you developed a questionnaire as part of this study and it is not under a copyright more restrictive than CC-BY, please include a copy, in both the original language and English, as Supporting Information.

Response: Thank you for the comments, we have revised the manuscript accordingly. For example, the survey questionnaires for this study consists of five components such as: socio-demographic characteristics, family-related questions, social support, psychotic symptoms, and the WHO-QOL tool questions. The sociodemographic and family related questions were developed from the previous literatures. Whereas the standard questionaries were used for the measurements of social support, psychotic symptoms, and health related quality of life.

We have submitted the survey questionary we used for this study as supplementary file. 

Q3. Thank you for stating the following financial disclosure:

Response: we update the financial disclosure in the revised manuscript and cover letter. 

Q4. In your Data Availability statement, you have not specified where the minimal data set underlying the results described in your manuscript can be found.

Response: in the revised manuscript we updated the Data Availability statement as “The datasets supporting the conclusions of this manuscript are available upon request to the corresponding author”. 

Reviewers' comments

Reviewer #1 Comments and Authors Response

Q1: please improve result section with apply and mention statistical.

Response: Thank you so much for your important comment. We have already revised the result in abstract section in the revised manuscript. 

Q2: How do they reach the conclusions in the abstract?

Response: Based the study results and by considering the gap, we tried to conclude our findings. 

Q3: please use update references in this section.

Response: Thank you so much for your important comment and for giving us the opportunity to improve the revised manuscript. We have revised the background section with updated reference based on your recommendation.

Q4: what are your hypotheses? please mentioned its in this section or methods section.

Response: Thank you again for asking these questions and giving us an opportunity to improve the revised manuscript. We have added the hypotheses of the study in the revised manuscript. 

The study hypothesis was: 

1) Null hypothesis = Substances user youths have good health-related quality of life 

Alternative hypothesis = Substances user youths have poor health-related quality of life

2) Alternative hypothesis = Health related quality of life is associated with socio-demographic, family related, and psychosocial factors 

Q5: what’s are response rate in your study?

Response: since our minimum sample size to run the SEM model was 370, we have invited 380 participants for our study. Three hundred seventy-two (372) youths were willing and interviewed the questionnaires with the response rate of 97.8%. 

Q6: sampling technique and study setting very vague. please explain more. Geographical cluster of each district, is setting of your study, or special centers of "Kebele" are your setting?

Response: sorry for this vague information. The special centers of Kebele are our study settings. The study employed multi-stage sampling technique. We stand from central Gondar zone and selected 6 districts, then number of Kebeles are also selected from these districts. There are also special centers “Ketenas” inside Kebele. Here we have recruited our participants from the Ketena’s in the form of cluster by considering a proportional allocated sample in each Kebeles.

Q7: Were these questioner adapted from an existing tested questionnaire or did the authors develop it themselves, tested it?

Response: Thank you again for asking these questions. The tools used in this study were Standardized and validated previously except the socio-demographic and some personal related variables. But the internal reliability of the tools was tested in our setting. 

Q8: Discussion does not explain the findings, Authors state other studies, but do not offer an explanation as to why they see the results that they see the results that they see

Response: Thank you again and for allowing us to improve the quality of the manuscript. We have already revised the discussion and updated the revised manuscript. Thanks again.

Q9: Please try to include more relevant literature

Response: Thank you again and for allowing us to improve the quality of the manuscript. We have added more literatures in the revised manuscript. 

Reviewer #2 Comments and Authors Response

Q1: The study sample size is reasonable with adequate rule of thumb for 370 subjects. The model appears to have reasonable goodness of fit and RMSEA values. The analysis software used, which is routine in this context, appeared to adequately identify factors associated with health-related quality of life which is a major factor, naturally. Structural equation modeling was used to estimate the relationships among exogenous, mediating, and endogenous variables which is notably the purpose of such software. There are many variables and the presentation is a descriptive summary of the relationships sought by the authors. One can get lost in the presentation with so much being described.

Response: Many thanks for the thoughtful comments and constructive suggestions, which help to improve the quality of this manuscript.

Q2: The characterization of the strength of the interdomain correlations is troubling. The authors note for example that Inter-domain correlation showed that there was a statistically significant correlation between domains, there is a highly positive correlation between environmental health domain and psychological health domain (r=0.61, p<0.001) and as compared with other domain's, psychological health domain and social relation health domain had a relatively

weak correlation between them (r= 0.49, p<0.001). How are weak and strong being decided?

Response: The value of correlation coefficient (r) is lies between -1 and 1 inclusively. The Interpretation of r is as follow, perfect positive linear relationship (if r= 1) b/n the variables, no linear relationship (if r = 0), perfect negative linear relationship (if r = -1). Many scholars are agreeing to classify any value between correlation coefficient into strong positive (1 to 0.5), weak positive (0.49 to 0.1), strong negative (-0.5 to -1) and weak negative (-0.1 to 0.49).

Q3: The format of Table 3 makes it difficult to read and interpret the table. Please correct the alignment. There are typos in the manuscript. For example, in the title of Fig. 3, Self Reted should obviously be Self Rated. Parcel in Figure 4 is in the footnote, but it looks like Parcil in the figure. 

Response: We thank the reviewer for the comment, we have gotten our mistakes, and we have corrected the comments in the manuscript. Thanks again.

Q4: The English should also be edited in places for clarity

Response: we are grateful for this comment which was our gap as it points to important issue to improve the readability of the text. After your suggestion, we have revised again the document, and then we have re-edited the grammatical error sentences by consuming more time and energy. We have also provided the document for professional language copy editor for advanced language edition. Based on this we have rewritten the whole document in more understandable to resolve language problems. Hence grammatical problems are resolved and changes/ modifications are highlighted by red color in the track changed file. 

Furthermore, we have modified, added, and changed a lot things from the topic to references based on editor and reviewers’ suggestions. Finally, we thank the editor and reviewers for your kind comments, constructive criticism and useful suggestions for which we have used to improve our manuscript.

---

## [Decision Letter · Decision Letter 1]

4 Sep 2022

Quality of life and associated factors among the youth with substance use in Northwest Ethiopia: Using Structural Equation Modelling

PONE-D-22-02286R1

Dear Dr. Tarekegn

We’re pleased to inform you that your manuscript has been judged scientifically suitable for publication and will be formally accepted for publication once it meets all outstanding technical requirements.

Kind regards,

Marianna Mazza

Academic Editor

PLOS ONE

Additional Editor Comments (optional):

Reviewers' comments:

Reviewer's Responses to Questions

**Comments to the Author**

1. If the authors have adequately addressed your comments raised in a previous round of review and you feel that this manuscript is now acceptable for publication, you may indicate that here to bypass the “Comments to the Author” section, enter your conflict of interest statement in the “Confidential to Editor” section, and submit your "Accept" recommendation.

Reviewer #1: All comments have been addressed

Reviewer #2: All comments have been addressed

2. Is the manuscript technically sound, and do the data support the conclusions?

Reviewer #1: Yes

Reviewer #2: (No Response)

3. Has the statistical analysis been performed appropriately and rigorously? 

Reviewer #1: Yes

Reviewer #2: (No Response)

4. Have the authors made all data underlying the findings in their manuscript fully available?

Reviewer #1: (No Response)

Reviewer #2: (No Response)

5. Is the manuscript presented in an intelligible fashion and written in standard English?

Reviewer #1: Yes

Reviewer #2: (No Response)

6. Review Comments to the Author

Reviewer #1: Thanks for responding,the authors have adequately addressed comments and in my opinion this manuscript is now acceptable for publication.

Reviewer #2: (No Response)

7. PLOS authors have the option to publish the peer review history of their article (what does this mean?). If published, this will include your full peer review and any attached files.

Reviewer #1: **Yes: **Dr.Hanieh Jormand

Reviewer #2: No

---

## [Editor Report · Acceptance letter]

9 Sep 2022

PONE-D-22-02286R1 

Quality of life and associated factors among the youth with substance use in Northwest Ethiopia: Using Structural Equation Modeling. 

Dear Dr. Tarekegn:

I'm pleased to inform you that your manuscript has been deemed suitable for publication in PLOS ONE. Congratulations! Your manuscript is now with our production department. 

Kind regards, 

on behalf of

Dr. Marianna Mazza 

Academic Editor

PLOS ONE